# Synergistic Mn-Co catalyst outperforms Pt on high-rate oxygen reduction for alkaline polymer electrolyte fuel cells

Ying Wang[1,2], Yao Yang [3], Shuangfeng Jia[4], Xiaoming Wang[5], Kangjie Lyu[1], Yanqiu Peng[1], He Zheng [4], Xing Wei[1], Huan Ren[1], Li Xiao[1], Jianbo Wang [2,4], David A. Muller [6], Héctor D. Abruña [3], Bing Joe Hwang [5], Juntao Lu[1] & Lin Zhuang [1,2]

Alkaline polymer electrolyte fuel cells are a class of fuel cells that enable the use of non-precious metal catalysts, particularly for the oxygen reduction reaction at the cathode. While there have been alternative materials exhibiting Pt-comparable activity in alkaline solutions, to the best of our knowledge none have outperformed Pt in fuel-cell tests. Here we report a Mn-Co spinel cathode that can deliver greater power, at high current densities, than a Pt cathode. The power density of the cell employing the Mn-Co cathode reaches 1.1 W cm$^{-2}$ at 2.5 A cm$^{-2}$ at 60 °C. Moreover, this catalyst outperforms Pt at low humidity. In-depth characterization reveals that the remarkable performance originates from synergistic effects where the Mn sites bind $O_2$ and the Co sites activate $H_2O$, so as to facilitate the proton-coupled electron transfer processes. Such an electrocatalytic synergy is pivotal to the high-rate oxygen reduction, particularly under water depletion/low humidity conditions.

[1] College of Chemistry and Molecular Sciences, Hubei Key Lab of Electrochemical Power Sources, Wuhan University, Wuhan 430072, China. [2] The Institute for Advanced Studies, Wuhan University, Wuhan 430072, China. [3] Department of Chemistry and Chemical Biology, Baker Lab, Cornell University, Ithaca, New York 14853, USA. [4] School of Physics and Technology, Center for Electron Microscopy, MOE Key Laboratory of Artificial Studies, Wuhan University, Wuhan 430072, China. [5] Department of Chemical Engineering, National Taiwan University of Science and Technology, Taipei 10607, Taiwan. [6] School of Applied and Engineering Physics, Cornell University, Ithaca, New York 14853, USA. Correspondence and requests for materials should be addressed to L.X. (email: chem.lily@whu.edu.cn) or to H.D.A. (email: hda1@cornell.edu) or to L.Z. (email: lzhuang@whu.edu.cn)

The recent decade has witnessed tremendous progress in both materials developments and catalysis studies of alkaline polymer electrolyte fuel cells (APEFCs)[1–9]. Research efforts have been driven by the fact that polymeric alkaline electrolytes can not only simplify the cell structure and operation, but also provide opportunities for employing non-precious metal catalysts[10–14]. However, despite great efforts, the last objective has remained elusive. While some materials, such as nitrogen-doped carbon-based materials[15,16], have been suggested to exhibit Pt-comparable activity towards the oxygen reduction reaction (ORR) in alkaline media, their performance is still much lower than that of Pt in APEFCs[17,18], especially when operated at high current densities necessary in automotive applications.

The screening of fuel-cell electrocatalysts is generally carried out using rotating disk electrode (RDE) voltammetry. However, the RDE experimental conditions are distinctly different from those in a polymer electrolyte fuel cell, where the electrode is fed with humidified gas, and the catalyst surface is under a humid atmosphere rather than in contact with an aqueous solution[19], as is the case under RDE conditions. Thus, it is not surprising that good-performing electrocatalysts in RDE tests can often exhibit poor performance under fuel-cell operation.

Here, we report an unexpected finding that the Mn-Co spinel catalyst (denoted hereafter as MCS) exhibits activity that is inferior to that of Pt, for ORR in RDE tests, but superior performance in APEFC tests, in particular under low-humidity conditions. At 60 ℃, the power density of APEFC employing such a MCS cathode reaches 1.1 W cm² at 100 relative humidity (RH%) and 0.92 W cm⁻² at 50 RH%, in comparison to 1 W cm² at 100 RH% and 0.67 W cm⁻² at 50 RH% for a Pt cathode. Through comprehensive characterizations, an unreported synergistic effect of the MCS surface is unraveled, where the Mn sites prefer $O_2$ binding and the Co sites favor $H_2O$ activation. Such a mechanism is pivotal in APEFC cathode, where water is a reactant but usually depleted.

## Results

**Electrochemical and fuel-cell tests**. Figure 1a presents typical RDE profiles for the ORR catalyzed by Pt and MCS in 1.0 M KOH solution. A negative shift of 50 mV in the half-wave potential clearly indicates that the ORR occurs at a lower rate on MCS than on Pt, and this trend does not change with potential as evidenced in the Tafel plots (inset to Fig. 1a). Such an observation would usually lead to the conclusion that the MCS would not be a good choice as ORR electrocatalyst for APEFCs. However, the fuel cell tests tell a different, and most unexpected, story (Fig. 1b). An APEFC with a Pt-Ru anode and a Pt cathode, exhibiting a peak power density (PPD) of 1 W cm⁻², is a benchmark of current APEFC research[20,21].

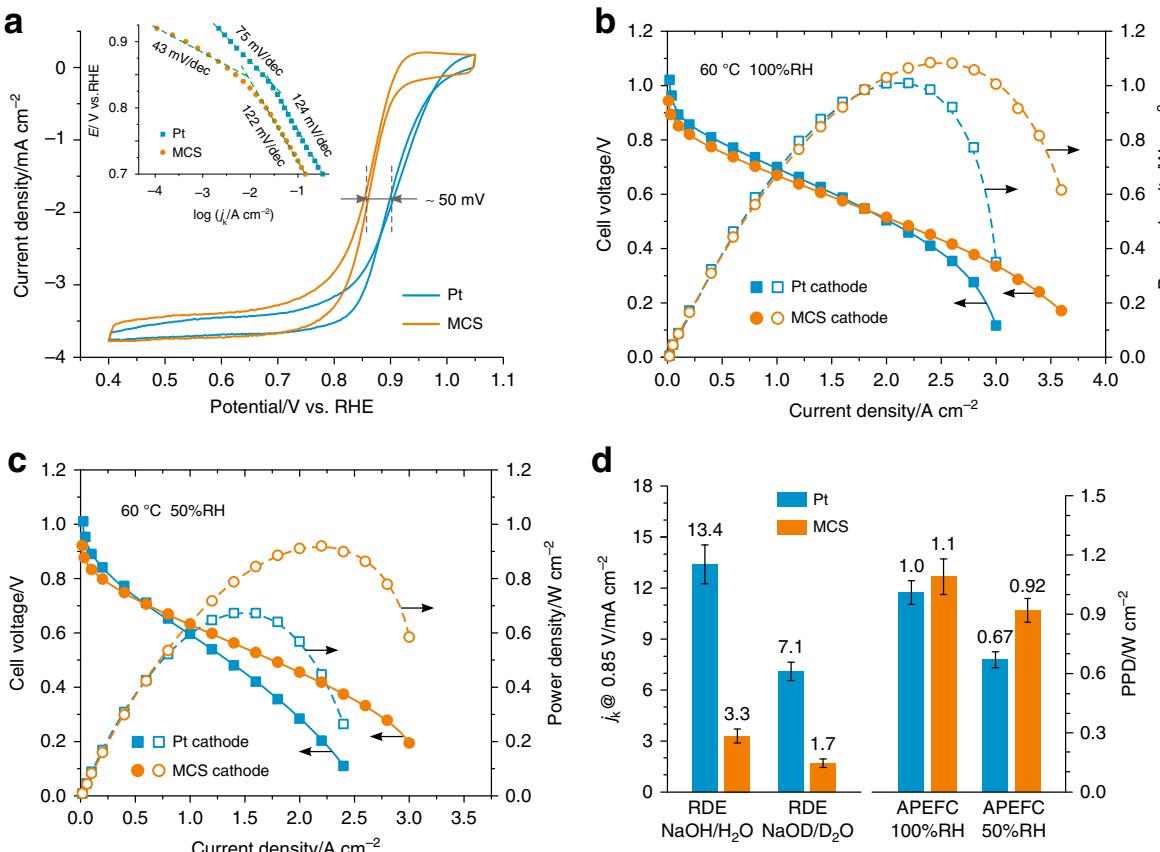

**Fig. 1** Comparison of Mn-Co spinel (MCS) catalyst and commercial Pt catalyst. **a** Rotating disk electrode (RDE) measurements in $O_2$-saturated KOH solution (1 mol L⁻¹) using 40 wt% Pt/C (Johnson Matthey, 50 μg$_{Pt}$ cm⁻²) and 40 wt% MCS/C (72 μg$_{metal}$ cm⁻²), respectively. Inset: Tafel plots. Scan rate = 5 mV s⁻¹. Rotation rate = 1600 rpm. See Supplementary Figs 1 and 2 for relevant electrochemical data. **b, c** Alkaline polymer electrolyte fuel cell (APEFC) tests with $H_2$ and $O_2$ at different relative humidities (RH). Anode catalyst: 60 wt% Pt-Ru/C (Johnson Matthey, 0.4 mg$_{metal}$ cm⁻²). Cathode catalyst: 60 wt% Pt/C (Johnson Matthey, 0.4 mg$_{Pt}$ cm⁻²) or 40 wt% MCS/C with an optimized loading of 0.58 mg$_{metal}$ cm⁻². (See Supplementary Fig. 3 for results with different catalyst loading.) Alkaline polymer electrolyte: aQAPS-S$_8$ membrane (35 μm in thickness) and aQAPS-S$_{14}$ ionomer (20 wt% in electrode)[4]. See Supplementary Figs 4 and 5 for impedance measurements and iR-corrected plots. Operation temperature = 60 °C. Backpressure = 0.1 MPa. **d** Performance comparison: Kinetic current densities ($j_k$) at 0.85 V, calculated from the RDE data recorded in 1 mol L⁻¹ NaOH/$H_2O$ and 1 mol L⁻¹ NaOD/$D_2O$ (See Fig. S6 for relevant results of isotopic labeling experiments), and the peak power density (PPD) resulting from APEFC tests

Upon replacing the Pt cathode with our MCS cathode, the cell performance underwent a slight loss at low current densities, but, as the current density increased, it kept increasing in a steady fashion, reaching a higher PPD of 1.1 W cm$^{-2}$, a performance metric never previously achieved in APEFCs with a non-precious metal cathode catalyst to the best of our knowledge. The MCS cathode can even sustain a current density of 3.5 A cm$^{-2}$, pointing to its inherently high activity.

Moreover, the MCS cathode dramatically outperforms the Pt cathode at low relative humidity over a wider range of current densities. As illustrated in Fig. 1c, when the humidity was lowered to 50 RH%, a significant drop in cell performance was observed for the Pt cathode, with the PPD decreasing by one third to 0.67 W cm$^{-2}$. However, for the MCS cathode, the PPD remained virtually unchanged at 0.92 W cm$^{-2}$. The ability to work at low relative humidity is a unique advantage for APEFC cathodes, where water (which is a reactant; *vide-infra*) is often depleted, particularly when the cell is operated at high current densities[22]. It should be noted that the Pt cathode has been well optimized to reach its maximum performance, the observed superiority of the MCS cathode, at high current densities and low humidity, is not due to a structural effect of the electrodes (including the thickness of the catalyst layer). In fact, the Pt cathode is thinner in the catalyst layer, which possesses lower electrical resistance than the MCS cathode (Supplementary Figs 4 and 5). Since the operation conditions (gas backpressure, flow rate, etc.) are the same for both electrodes, the mass transport should not be particular to the thinner Pt cathode. The obvious difference in the water/humidity dependence of the cathode performance has to be related to a certain catalyst-water interaction.

Figure 1d summarizes the activity comparison between Pt and MCS under different experimental conditions in RDE and fuel-cell tests. While Pt is superior, over the MCS, under water-rich conditions, it becomes inferior at low water content. This suggests the presence of an effect, on the ORR catalytic activity, that is sensitive to the water content and works oppositely on Pt and MCS. In APEFCs, $H_2O$ is not only necessary for ionic conduction, but is also a reactant in the ORR (Eq. 1).

$$O_2 + 2H_2O + 4e^- \leftrightarrows 4OH^- \qquad (1)$$

Proton transfer processes in this reaction are as crucial as the electron transfer events themselves[23,24], as evidenced (via H/D isotope effects) by the significant diminution of the kinetics of the ORR in $NaOD/D_2O$ solution (Fig. 1d and Supplementary Fig. 6). Thus, the ORR will be highly sensitive to the amount, and state, of $H_2O$ just above the catalyst surface[25,26], especially when $H_2O$ is a minority species in the gaseous phase. The high catalytic activity of MCS toward the ORR at low $H_2O$ content suggests the presence of a special affinity for $H_2O$, in addition to the appropriate interactions with $O_2$.

**Material characterization.** In an effort to unveil the origin of these effects, in-depth characterization of the structure and surface properties was carried out. Synchrotron X-ray diffraction (XRD, Fig. 2a) clearly indicates the presence of the cubic spinel structure $(AB_2O_4)$[27] with a lattice constant $a = 8.2938$ Å. The formal valence of Mn and Co in the MCS sample exhibiting optimal ORR performance (nominally $Mn_{1.5}Co_{1.5}O_4$) were analyzed using X-ray absorption near-edge structure (XANES, Fig. 2b), which yielded values of +2.76 and +2.56, respectively. The stoichiometry of the spinel $(AB_2O_4)$ was thus determined to

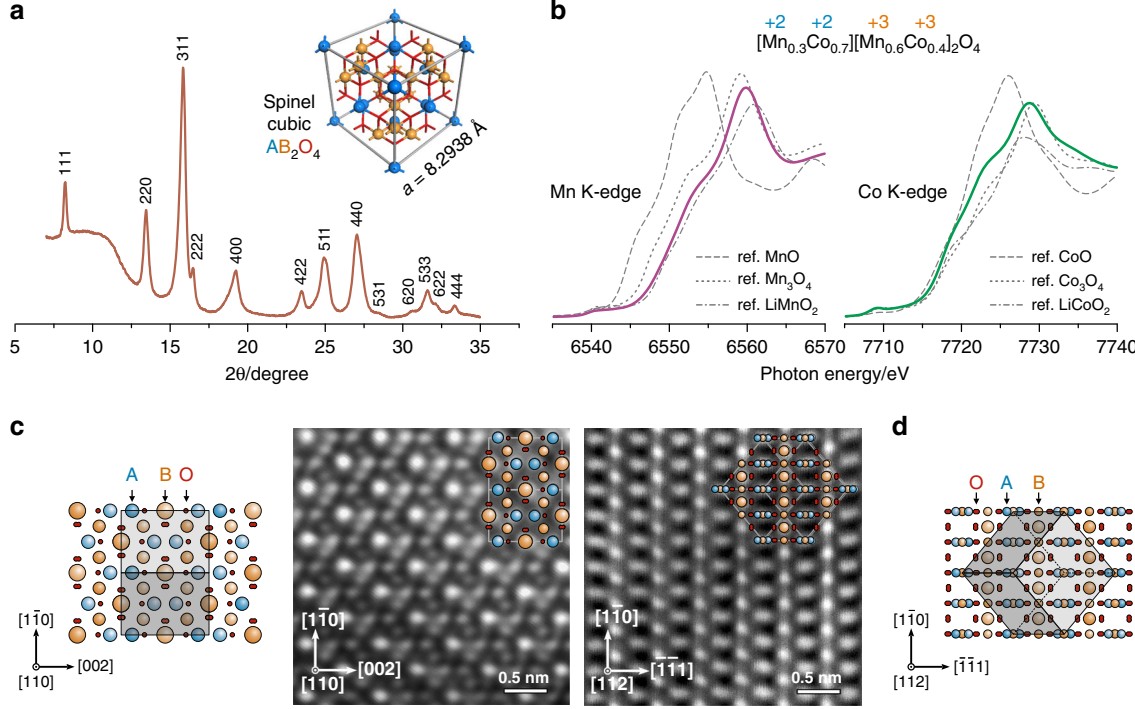

**Fig. 2** Structural characterizations of the Mn-Co catalyst. **a** Synchrotron X-ray diffraction (XRD) pattern, identifying the spinel cubic crystal structure with a lattice constant $a = 8.2938$ Å (inset). X-ray wavelength $\lambda = 0.68876$ Å. The raised baseline at around 10° is due to the carbon black support of the Mn-Co spinel (MCS). **b** X-ray absorption near-edge structure (XANES) spectra. The K-edge absorptions of Mn and Co were collected, each with three reference samples. The formal valences of Mn and Co were determined to be +2.76 and +2.56, respectively, corresponding to a stoichiometry of $[Mn_{0.3}Co_{0.7}][Mn_{0.6}Co_{0.4}]_2O_4$. **c, d** High-angle annular dark-field images from scanning transmission electron microscopy (HAADF-STEM) images of the MCS lattice, taken on zone axes of [110] and [112]. Models of lattice projection are provided, with a unit cell embedded in the picture, to interpret the atomic resolution images. See Supplementary Figs 8 and 9 for reasoning of the spot brightness. Elemental mapping results are provided in Supplementary Fig. 10

be $[Mn_{0.3}Co_{0.7}][Mn_{0.6}Co_{0.4}]_2O_4$, indicating that while Co is distributed almost uniformly at the tetrahedral (A) and octahedral (B) sites of the spinel lattice, Mn is enriched at the B site. The elemental ratio of Mn/Co is 3/2 at the catalytically active B site[28]. Scanning transmission electron microscopy (STEM) observations indicated that the MCS particles are irregularly-shaped nanocrystals (Supplementary Fig. 7). The high-angle annular dark-field STEM images (HAADF-STEM, Fig. 2c, d), taken on the [110] and [112] zone axes, provide atomic views of the arrangement of metal ions inside the MCS lattice. The high-contrast patterns match well the lattice model reconstructed based on the above-determined stoichiometry (see Supplementary Figs 8 and 9 for detailed interpretation). In addition, the atomic-scale elemental mapping, using energy-dispersive X-ray spectroscopy (Supplementary Fig. 10), also confirms the enrichment of Mn at the B sites.

**Surface analyses**. In an attempt to distinguish the functionality of the Mn sites and Co sites on the spinel surface, we deliberately prepared MCS samples with Mn-segregated and Co-segregated surfaces, denoted as Mn-MCS and Co-MCS, respectively. The success in controlling surface segregation was ascertained by elemental mapping using electron energy loss spectroscopy (EELS, insets of Fig. 3a and Supplementary Figs 11–13). Samples were then characterized with X-ray photoelectron spectroscopy (XPS) to identify the oxygen-containing surface species (Fig. 3a). In addition to $O^{2-}$ that constitutes the spinel, $OH_{ads}$ and $H_2O_{ads}$

were identified by their distinct chemical shifts[29,30]. While Mn segregation resulted in an enhancement in the $O^{2-}$ component and a diminution of $H_2O_{ads}$, Co segregation caused a reversal effect with a clear increase in the $H_2O_{ads}$ component. These results suggest that the actual MCS surface consists mainly of Mn-OH/Mn-O and Co-OH/Co-OH$_2$, in agreement with the zeta-potential analysis in solutions of varying pH. As shown in Fig. 3b, the potential of zero charge (PZC) of MCS appears at pH = 5.5, and shifts to pH = 8.5 upon Co segregation, and to pH = 3.3 upon Mn segregation. These observations suggest that the Co sites interact weakly with O, and tend to be positively charged, likely as Co-OH$_2^{\delta+}$, while the Mn sites have a strong affinity for O, and tend to be negatively charged, likely as Mn-O$^{\delta-}$.

The above experimental observations of the surface character of the MCS are in qualitative agreement with density functional theory (DFT) calculations (Fig. 3c). The Mn sites on MCS are highly active for binding both $O_2$ and $H_2O$, but the Mn-$O_2$ interaction is stronger than the Mn-OH$_2$ interaction, suggesting that the Mn sites prefer to bind $O_2$ over $H_2O$ when the MCS is exposed to humid air. In contrast, the Co sites have a notable affinity for $H_2O$ but appear not to bind $O_2$ by itself (given that $O_2$ can adsorb at the bridge sites between Mn and Co atoms, see Supplementary Table 1). Hence, when the MCS cathode is exposed to humid $O_2$, the Mn sites and Co sites on the surface bind different adsorbates, preferentially yielding Mn-$O_2$ and Co-OH$_2$, respectively.

The DFT calculations can also provide an assessment of the capability of breaking the O-O bond at a catalytic site.

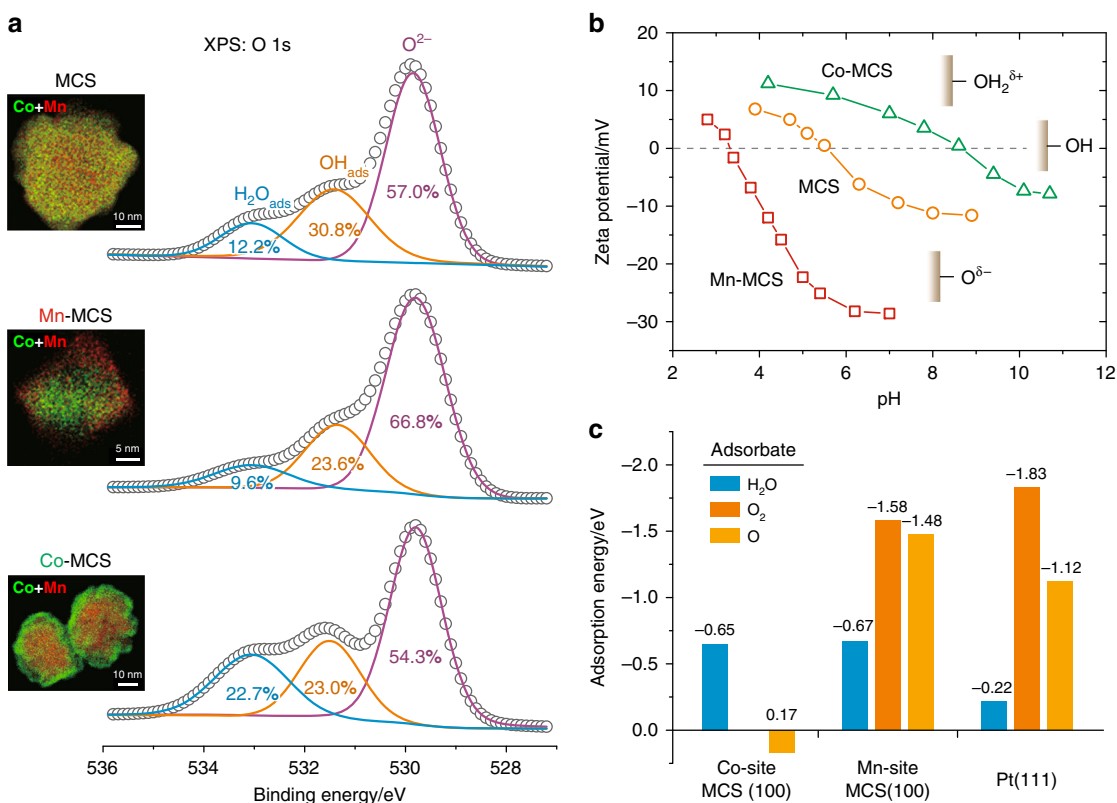

**Fig. 3** Surface analyses of the Mn-Co catalyst. **a** Oxygen 1 s spectra of X-ray photoelectron spectroscopy (XPS) for MCS and two reference samples with Mn or Co enriched on the surface (denoted as Mn-MCS and Co-MCS, respectively). Insets are electron energy loss spectroscopy (EELS) mapping for these samples (also see Supplementary Figs 11–13). Spectral deconvolution identified three distinct chemical environments of O, corresponding to those of $H_2O_{ads}$, $OH_{ads}$, and $O^{2-}$ [29,30]. **b** Zeta-potential measurements for MCS, Mn-MCS, and Co-MCS particles dispersed in solutions of different pH. **c** Density functional theory (DFT) calculated adsorption energies for $H_2O$, $O_2$, and O on the Mn and Co sites of the MCS (100) surface, in comparison to those on Pt (111). No stable adsorption structure was found for $O_2$ on the Co site of MCS (100). See Supplementary Tables 1–6 for Supplementary Data of DFT calculations. The adsorption energy of O was defined relative to half the energy of $O_2$, such that negative values indicate a spontaneous dissociation of $O_2$ on the surface

Specifically, the adsorption energy of an O atom is calculated relative to half the energy of $O_2$, such that negative values designate a thermodynamically spontaneous dissociation of $O_2$. As presented in Fig. 3c, the dissociation of $O_2$ is energetically highly favorable on the Mn site, but disfavored (albeit slightly) on the Co site. One can thus conclude that the MCS possesses a synergistic surface for ORR catalysis, with the Mn sites binding and cleaving $O_2$, and the Co sites enriching and activating $H_2O$, so as to facilitate the proton-coupled electron transfer processes of oxygen reduction.

**Water activation in oxygen reduction.** The proposed synergic mechanism of the MCS-catalyzed ORR is illustrated, stepwise, in Fig. 4a. Assuming that Mn-OH and Co-OH represent the initial states, the $O_2$ is preferentially bound to the Mn site to yield Mn-$O_2$, along with a $1e^-$ reduction to produce $OH^-$. $H_2O$ is preferentially bound to the Co site, as Co-$OH_2$, also with a $1e^-$ reduction to generate $OH^-$. A surface proton transfer (reaction I) can then occur from the Co-$OH_2$ to the proximate Mn-$O_2$, leading to a regenerated Co-OH and a Mn-OOH species that is followed by a $1e^-$ reduction to produce Mn=O and $OH^-$. The Mn=O can take the second proton, transferred from Co-$OH_2$, to regenerate the Mn-OH (reaction II). The extraordinary feature of this mechanism includes the proton mediation by the turnover of Co-OH/Co-$OH_2$ and the surface proton transfer between the proximate Co and Mn sites. On the one hand, based on DFT calculations, the O–H bond energy decreases from 5.14 eV to

3.42 eV when the $H_2O$ is bound to the Co site (upper-right inset of Fig. 4a). On the other hand, the energy barriers for reaction I and II are small (central inset of Fig. 4a, Supplementary Tables 7 and 8). We believe that these energetic features are fundamental to the $H_2O$ activation and proton-transferred reduction of $O_2$ on the MCS surface.

To ascertain the involvement of surface $H_2O$ in the MCS-catalyzed ORR, in-situ attenuated total reflection Fourier transform infrared spectroscopy (ATR-FTIR) was employed to detect the subtle changes in the $H_2O$ vibrations on Pt and MCS surfaces under electrochemical conditions (Fig. 4b). The $H_2O$ bending vibration, $\delta(HOH)$, turns out to be at higher wavenumbers on Pt than on MCS, indicating that the Pt-$H_2O$ interaction is weaker than the MCS-$H_2O$ interaction[31]. Even more compelling evidence for the strong adsorption of $H_2O$ on the MCS surface is provided by the Stark effect, namely, a significant potential-dependent shift in the $\delta(HOH)$ wavenumber of 25 cm$^{-1}$ V$^{-1}$. In contrast, such a Stark effect is negligible on a Pt surface. These experimental observations are consistent with the DFT calculations (Fig. 3c) that show that the adsorption energy of $H_2O$ on Pt (111) is only one-third of that on MCS (100).

Additional valuable information was provided by in-situ ATR-FTIR experiments, showing that the $\delta(HOH)$ wavenumber decreased on both Pt and MCS when the atmosphere was switched from Ar to $O_2$ (Fig. 4b). This can only be ascribed to the additional interaction between the surface $H_2O$ and the surface oxygen species, providing unambiguous evidence for the

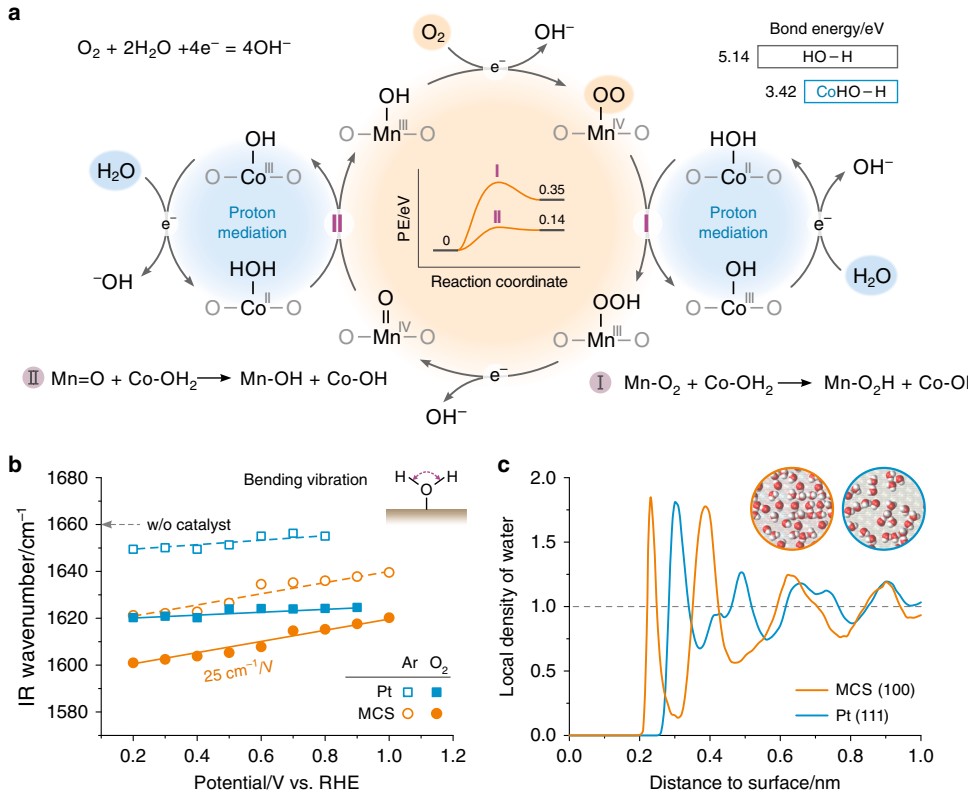

**Fig. 4** Reaction mechanism involving water activation. **a** Schematic illustration of the proposed synergistic mechanism of ORR on MCS, featuring the dissociative reduction of $O_2$ at the Mn site, the proton mediation by the Co site, and the surface proton transfer in between (reactions I and II). Inset central: DFT-calculated energy barriers for reactions I & II on MCS (100) (See Supplementary Tables 7 and 8 for details). Inset upper-right: DFT-calculated bond energies of O–H in $H_2O$ and Co-$OH_2$. **b** Results of in situ attenuated total reflection Fourier transform infrared (ATR-FTIR) studies for MCS and Pt electrodes in Ar or $O_2$ saturated KOH solutions. See Supplementary Fig. 14 for relevant FTIR spectra. The IR signals of interest were from the bending vibration of $H_2O$ (inset). The Stark effect (wavenumber shift with potential) is a measure of the $H_2O$ adsorption on the surface. **c** Local density of water on MCS (100) and Pt (111) surfaces at 300 K, obtained from atomistic molecular dynamics (MD) simulations (also see Supplementary Fig. 15). Inset: Snapshots of water molecules in a surface layer up to 0.3 nm thick

involvement of surface $H_2O$ in the ORR. Moreover, the $\delta(HOH)$ Stark effect remained unchanged on MCS during the ORR, indicating that the surface $H_2O$ has not been repelled by the co-adsorption of $O_2$; whereas the originally weak Stark effect of $\delta(HOH)$ on Pt could barely be observed during the ORR, suggesting that the $H_2O$ is likely to be further away from the Pt surface when covered by oxygen species. The identification of the different functionalities of the Mn sites and Co sites, and the direct observation of the involvement of surface $H_2O$ in the MCS-catalyzed ORR, provide strong support for the synergistic mechanism (Fig. 4a).

The above computational and experimental observations clearly establish the superior activity of MCS over Pt for the ORR under conditions of high current density and low humidity. The lower ORR activity of MCS in RDE tests is also under-standable. In $O_2$-saturated KOH solutions, the molar ratio of $H_2O/O_2$ is over $10^4$, so that the hydrophilic MCS surface is dominated by $H_2O$[32,33], despite the oxophilicity of the Mn sites. Molecular dynamics (MD) simulations (Fig. 4c) show that liquid water can wet the MCS surface with a proximity of 0.23 nm, in comparison to 0.3 nm on a Pt surface. Thus, there is more space for $O_2$ adsorption on a Pt surface than on an MCS surface under water-flooding conditions.

## Discussion

Although MCS-like materials were used as ORR catalysts in the literature[12–14], the mechanistic understanding has been vague and the cell performance observed here is unprecedented to the best of our knowledge. As for the stability, the MCS catalyst turns out to be more stable than Pt under the test protocol of potential cycling (Supplementary Fig. 16). Our findings represent not only the discovery of a practical, high-performance non-precious metal catalyst for APEFCs, but also reveal a strategy for the ORR catalyst design. In addition to the electronic effects that have often been used and/or invoked to tune the reactivity of solid surfaces toward $O_2$[34,35], designing synergistic surfaces that can activate $H_2O$ and facilitate proton transfer processes is also pivotal for ORR catalysts, in particular for metal oxides working in alkaline media where both $O_2$ and $H_2O$ are reactants.

## Methods

**Materials**. Cobalt(II) acetate (Co(OAc)$_2$·4H$_2$O, 99.0%), manganese(II) acetate (Mn(OAc)$_2$·4H$_2$O, 99.0%), ammonium hydroxide (NH$_3$·H$_2$O, 25%-28%), potassium hydroxide (KOH, 99.99%), sodium hydroxide (NaOH, 99.99%) were purchased from China Medicine Shanghai Chemical Reagent Corporation. Sodium deuter-oxide (40 wt% in D$_2$O, 99 atom% D), deuterium oxide (D$_2$O, 99.9 atom% D) were obtained from Sigma-Aldrich and used without further purification. Vulcan XC-72 (Cabot Co.), 60 wt% Pt/C (Johnson Matthey Co.), 60 wt% PtRu/C (Johnson Matthey Co.), Nafion (Dupont, 5%) were used as received. The alkaline polymer electrolytes ($a$QAPS-S$_x$) were prepared as reported in our previous work[4].

**Catalyst preparation**. In a typical synthesis of Mn-Co spinel (MCS) catalyst supported on carbon black, Co(OAc)$_2$·4H$_2$O (63.5 mg) and Vulcan XC-72 (pre-heated at 110 °C in air, 60.0 mg) were added to ultrapure water (30 mL). After ultrasonic blending for 15 min, 0.5 mL of NH$_3$·H$_2$O was added under magnetic stirring, followed by the addition of Mn(OAc)$_2$·4H$_2$O aqueous solution (62.5 mg dissolved in 5 mL water). The mixture was aged at a controlled temperature under magnetic stirring for 2 h. After that, the suspension was ultrasonically blended for 10 min, and then transferred to a 40 mL Teflon autoclave for hydrothermal reac-tion at 150 °C for 3 h. The resulting product was collected by centrifugation and washed with water, then freeze dried under vacuum. To ensure a homogeneous distribution of Co and Mn in MCS, the ageing temperature must be set to 60 °C. To prepare MCS with Mn segregation on the surface (denoted as Mn-MCS), the ageing temperature was set to 40 °C; while to prepare Co-segregated samples (Co-MCS), the heating was set to 80 °C before the addition of Mn(OAc)$_2$ aqueous solution. MCS samples without carbon black support were synthesized employing the same procedure without the addition of Vulcan XC-72.

**Electrochemical measurements**. To prepare the working electrode, a 5 mg sample was dispersed ultrasonically in 1 mL of diluted Nafion alcohol solution (0.05 wt%)

to form an ink, and the suspension was pipetted onto a rotating ring-disk electrode (RRDE, $\phi$ = 4.57 mm), which was buff-polished with an alumina suspension ($\phi$ = 0.05 μm) prior to use. The catalyst coated electrode was dried under an infrared lamp. $a$QAPS ionomer was also tested as the binder, no difference was found from using Nafion ionomer.

Electrochemical experiments were conducted on a CHI-700A potentiostat with an RRDE system (Pine Research Instruments). A sheet of carbon paper (Toray) was used as the counter electrode. Hg/HgO in the same solution was used as the reference electrode, and all potentials were converted to the reversible hydrogen electrode (RHE) in the same solution. The oxygen reduction reaction (ORR) evaluation was carried out in O$_2$-saturated 1.0 M KOH solution. The rotation rate was 1600 rpm, and the ORR curves were recorded by cycling the potential between 0.40 V and 1.05 V (vs. RHE) at 5 mV s$^{-1}$. The kinetic current densities and Tafel plots were calculated by the average current densities of positive and negative scans. Cyclic voltammetry (CV) was carried out in Ar-saturated 1.0 M KOH solution by cycling the potential between 0.10 V and 1.40 V (vs. RHE) at different scan rates. All measurements were carried out at 25 °C.

The kinetic current densities ($j_k$) were calculated via Koutecky-Levich analysis:

$$\frac{1}{j} = \frac{1}{j_k} + \frac{1}{j_d}$$

where $j$ is the total current density, and $j_d = 0.62nFD_{O_2}^{2/3}\omega^{1/2}\nu^{-1/6}C_{O_2}^*$ is the diffusion limiting current density ($n$ = electrons transferred number, $F$ = Faraday's constant, $D_{O_2}$ = diffusion coefficient of O$_2$, $\omega$ = electrode rotation rate, $\nu$ = kinematic viscosity, $C_{O_2}^*$ = the bulk concentration of O$_2$).

To evaluate the H$_2$O$_2$ yield and the electron transfer number of the catalysts, the Pt ring potential was set to 1.3 V (vs. RHE) in the RRDE measurements. The H$_2$O$_2$ yield was calculated using the following equation[36]:

$$H_2O_2(\%) = 200 \times \frac{I_R/N_0}{(I_R/N_0) + I_D}$$

Where $I_R$ and $I_D$ are the ring and disk currents, respectively, and $N_0$ = 0.22 is the collection efficiency of the RRDE.

**Isotopic labeling experiments**. To study the kinetic isotope effect (KIE), 1.0 M NaOH and 1.0 M NaOD solutions were prepared with Milli-Q water (>18 MΩ cm) and D$_2$O, respectively. All vessels were dried under an infrared lamp before used, and the catalyst coated electrodes were soaked in the electrolyte solution for 30 min. before electrochemical experiments. The electrochemical experiments were conducted using the same procedure described above.

The KIE is the ratio of the rate constants ($k_f^H/k_f^D$), which can be calculated at a selected potential[37]

$$KIE = \frac{k_f^H}{k_f^D} = \frac{j_k^H n^D FC_{O_2}^{*,D}}{j_k^D n^H FC_{O_2}^{*,H}}$$

where

$$\frac{C_{O_2}^{*,D}}{C_{O_2}^{*,H}} = 1.101 \quad (298\,K)$$

due to the difference in the solubility of O$_2$ in D$_2$O and H$_2$O. By assuming that the electron count (number of electrons transferred) ($n^D = n^H$), the KIE can then be calculated as

$$KIE = \frac{k_f^H}{k_f^D} = \frac{j_k^H C_{O_2}^{*,D}}{j_k^D C_{O_2}^{*,H}} = 1.101\frac{j_k^H}{j_k^D}$$

The ORR KIEs on Pt and MCS at +0.85 V (vs. RHE) were calculated to be 2.08 ± 0.2 and 2.14 ± 0.2, respectively, both belonging to a primary isotope effect[38].

**Fuel cell tests**. The procedures for fabricating membrane-electrode assemblies and fuel cell tests were the same as reported previously[20]. Specifically, the catalyst powder was dispersed in a $a$QAPS-S$_{14}$ (Cl$^-$ as anion) ionomer solution and sprayed onto each side of an $a$QAPS-S$_8$ (Cl$^-$ as anion) membrane (35±5 μm in thickness) to fabricate the catalyst coated membrane (CCM). The area of electrodes was 4 cm$^2$. The catalyst loadings for PtRu/C in the anode and Pt/C in the cathode was 0.4 mg$_{metal}$ cm$^{-2}$. Different MCS loadings in the cathode were tested to find the optimal value. The weight percentage of $a$QAPS-S$_{14}$ in the electrodes was controlled to be 20 wt%. To replace the Cl$^-$ anion in the CCM to OH$^-$, the CCM was immersed in 1.0 M KOH solution for 10 h. Finally, the CCM in the OH$^-$ form, was repeatedly rinsed with deionized water until the pH of the residual water was neutral. It was then pressed between two pieces of carbon paper (AvCard GDS3250) to make the membrane-electrode assembly (MEA).

The H$_2$-O$_2$ fuel cells were tested (850e Multi Range, Scribner Associates Co.) under galvanic mode using humidified H$_2$ and O$_2$ gases. The cell temperature was set to 60 °C, and the flow rate of both H$_2$ and O$_2$ gases was 200 mL/min with 0.1 MPa of backpressure. To obtain 50 RH% of H$_2$ and O$_2$, the temperature of the humidifying water tanks was set as 45.7 °C.

**Synchrotron X-ray measurements**. Synchrotron X-ray diffraction (XRD) and X-ray absorption spectroscopy (XAS) measurements were conducted at the Taiwan Beam Line of BL12B1 in the Spring-8. The electron-storage ring was operated at 8 GeV with a current of 100 mA. For the XRD measurements, the MCS/C samples were placed in a glass capillary and sealed with a resin. The wavelength of the incident X-ray beam was set to 0.68876 Å, which is calibrated with a $CeO_2$ standard. XRD data were recorded on an imaging plate for 30 min over the range $2\theta = 0–40^o$ with a $2\theta$ step of $0.01^o$.

For the XAS measurements, a Si (111) double-crystal monochromator was employed for the energy selection with a resolution d$E/E$ better than $2 \times 10^{-4}$ at elemental edges. All XAS spectra were recorded at room temperature in transmission mode. Higher harmonics were eliminated by detuning the double-crystal Si (111) monochromator. Three gas-filled ionization chambers were used in series to measure the intensities of the incident beam ($I_o$), the beam transmitted by the sample ($I_t$), and the beam subsequently transmitted by the reference foil ($I_r$). The third ion chamber was used in conjunction with the reference metal foil for the elemental edge measurements. All measurements were compared against the reference samples. The X-ray absorption near-edge structure (XANES) was fitted by a linear combination with the spectra of standards, within a range of $-20$ to 30 eV of the $E_0$-normalized spectra, using the Athena software[39]. The oxidation state of Mn was fitted by a linear combination of MnO, $Mn_3O_4$, and $LiMnO_2$, while the oxidation state of Co was fitted by a linear combination of CoO, $Co_3O_4$, and $LiCoO_2$.

**Electron microscopy analysis**. Scanning transmission electron microscopy (STEM) imaging and energy-dispersive X-ray spectroscopy (EDX) mapping were acquired on a JEOL JEM-ARM200CF microscope operated at 80 kV with a Schottky cold-field emission gun in Wuhan University. The high angle annular dark field (HAADF) images were acquired with a beam convergence angle of 31 mrad. The EDX elemental mapping was carried using the JEOL SDD-detector with two 100 $mm^2$ X-ray sensor. The STEM imaging with electron energy loss spectroscopy (EELS) elemental mapping was acquired on a fifth-order aberration-corrected FEI Titan Themis operated at 60 keV at Cornell University. Atomic-scale EELS elemental mapping was acquired on a fifth-order aberration-corrected Nion UltraSTEM operated at 60 keV. Beam damage of sample was routinely examined before and after EELS mapping.

**Element and surface analyses**. Inductively coupled plasma optical emission spectrometry ICP-OES (IRIS Intrepid II XSP, Thermo) was used to determine the Mn and Co contents in the MCS samples. The X-ray photoelectron spectroscopy (XPS) data were collected on a Thermo Fisher EscaLab 250Xi spectrometer using Al Kα radiation ($h = 1486.6$ eV). The O 1 s spectra were fitted using the XPSPEAK41 software. Zeta-potential measurements were carried out on Malvern Zetasizer Nano ZSP equipped with a pH autotitrator. 10 mg MCS samples without carbon supporters were dispersed in 100 mL KCl solution (1 mmol $L^{-1}$) to obtain stable stock solutions for the subsequent measurements.

**In-situ infrared spectroscopy**. A Au film was prepared on the reflecting plane of an attenuated total reflection (ATR) Si prism by a secondary chemical deposition technique[40]. The thin Au film exhibited a surface-enhanced infrared absorption (SEIRA), which is very sensitive to the detection of surface absorbed species[41]. The catalyst ink was pipetted onto the Au film, dried under an infrared lamp. The MCS and Pt loadings were 50 μg/$cm^2$. The IR beam traveled through the Si prism with an incident angle of 70° to detect the surface absorption species on the working electrode by an evanescent infrared wave and finally arrived at the detector through the total reflection. An FTIR spectrometer (Thermo Fisher Nicolet 6700) equipped with an MCT detector was used for ATR-SEIRAS measurements. The spectral resolution was set to 8 $cm^{-1}$ and 64 interferograms were co-added for each spectrum. All IR spectra are presented in absorbance units, defined as $-\log (I/I_0)$, where $I$ and $I_0$ represent the spectral intensities of the sample and reference states, respectively. The reference spectra were collected at 1.2 V (vs. RHE) under Ar or $O_2$ atmosphere.

**Density functional calculations**. Density functional theory (DFT) calculations were performed using the Vienna Atomic Simulation Package (VASP, version 5.3)[42]. Generalized gradient approximation (GGA) of Perdew-Burke-Ernzerhof (PBE) was supplemented by the rotationally invariant "+U" description[43]. A projected augmented wave (PAW) basis, along with a plane-wave kinetic energy cutoff of 400 eV, was used throughout. The Hubbard U values of Mn and Co atoms were chosen as 4.4 eV and 5.4 eV, respectively[44]. Spin polarization calculations were carried out for all possible structures. For the calculation of surface and chemisorption systems, the spinel (100) surface was constructed as an eight-layer atom slab and repeated in super cell geometry with successive slabs separated by a vacuum region (10 Å). During the geometry optimization, the adsorbate layer and the top four layers of the slab were allowed to relax. For the calculations of surface and chemisorption systems of Pt, a Pt (111) surface was constructed as a four-layer metal slab. During geometry optimization, the adsorbate layer and the top two layers of the slab were allowed to relax. The energies were converged to 1 meV per atom and ionic relaxations were allowed until the absolute value of the force on

each atom was below 0.02 eV $Å^{-1}$. Minimum energy pathways (MEPs) were obtained by the climbing image nudged elastic band (CI-NEB) method with a force convergence to 0.03 eV $Å^{-1}$.

**Molecular dynamics simulations**. Classical atomistic molecular dynamics (MD) simulations were carried out using Gromacs[45]. The simulation was carried out in the NVT ensemble. The temperature was kept at 300 K using the Nosé-Hoover thermostat[46]. The cut-off for short-range interactions was 1.2 nm and the particle-mesh Ewald method was used to account for long-range electrostatics. A time step of 2 fs was used for the simulation. We carried out a 5 ns simulation with the coordinates saved every 1 ps.

The surfaces of Pt(111) and MCS(100) were directly taken from the equilibrium structures in DFT calculations. For the Pt surface, a simple Lennard-Jones potential was used. This potential has been widely used in various systems with water[47]. For the MCS surface, the universal force field was used[48]. The Chargemol program[49] was used to describe the DDEC6 atom charges, which can reproduce the electrostatic potential from DFT simulations. For water, we used the SPC/E model[50]. The Lorentz-Berthelot mixing rules were invoked for all interspecies interactions.

The box had a width and length of 4.7 and 6.2 nm, respectively, to match the lattice constants of Pt and MCS surfaces. The two surfaces were separated by a 10-nm water layer and fixed in their initial positions. The height of the box was chosen to be 15 nm so that a wide vacuum layer was generated on the other side to avoid influences between the two surfaces.

The radial distribution function was calculated for the oxygen atom in water around the metal atoms in the surface of Pt or MCS.

## Data availability

The source data underlying Figs 1, 2a, b, 3a, b and 4b, c are provided as a Source Data File. All relevant data are also available from the authors.

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

## Acknowledgements

This work was financially supported by the National Key Research and Development Program of China (2016YFB0101203) and the National Natural Science Foundation of China (91545205, 21872108). This work made use of TEM facilities of the Cornell Center for Materials Research (CCMR), which are supported through the National Science Foundation Materials Research Science and Engineering Center (NSF MRSEC) program (DMR-1719875), and the Center for Alkaline Based Energy Solutions (CABES) funded by the U.S. Department of Energy under Award DE-SC0019445.

## Author contributions

Y.W. performed electrochemical experiments and DFT calculations; Y.Y., S.J. and H.Z. performed TEM experiments; Y.Y. and X.-m.W. performed synchrotron X-ray experiments; K.L. performed MD simulations; Y.P. performed polyelectrolyte synthesis; X.W. performed FTIR measurements; H.R. performed complementary experiments; J.W., D.A.M., B.J.H., and J.L. performed the analysis of experimental data; L.X., H.D.A., and L.Z. supervised the whole work and wrote the paper.

## Additional information

**Competing interests:** The authors declare no competing interests.

