## [Peer Review File · Nature Communications]

Reviewers' comments:

Reviewer #1 (Remarks to the Author):

Review form for manuscript NCOMMS-18-33282-T "Synergistic Mn-Co Catalyst Outperforms Pt on High-Rate Oxygen 3 Reduction for Alkaline Polymer Electrolyte Fuel Cells", by Ying Wang et al.

The paper is a very comprehensive study of carbon-supported mixed Co and Mn oxides (MCS/C) for the alkaline oxygen reduction reaction, for potential use (and validation) in Alkaline polymer electrolyte fuel cells (APEFCs). The materials demonstrate a clear interest, as their performance is matching (or even surpassing) those of a Pt/C benchmark. The experimental data contains thorough physicochemical characterizations of the materials, as well as theoretical calculations by DFT and molecular dynamics. This is a very nice set of data, that would deserve publication, provided the following questions are adequately taken care of by the authors.

Major comments:

1- In RDE, the binder/ionomer was Nafion, whereas in APEFC it was the alkaline ionomer (aQAPS-S14); why did the authors make this choice? Could it influence the observed performances? For example, it starts to be observed that alkaline ionomers act as "poisons" of Pt electrocatalysts, while Nafion does less; if so, would the authors not "artificially" make their Pt/C electrode better in RDE than in APEFC? Please provide as well RDE data using the same alkaline ionomer for the two materials (Pt/C and MCS/C) to enable a fairer comparison between RDE and APEFC.

2- The results can also be biased because the active layers (both in RDE and in APEFC configuration) have not the same thicknesses (the MCS percentage on carbon is 40 wt% versus 60 wt% for Pt/C, and the loading at the RDE/APEFC cathode is larger than for Pt/C by 50%, which means that the active layer is roughly twice thicker for MCS/C than for Pt/C in all conditions). This can strongly influence the results. This is already witnessed in RRDE, where the H₂O₂ yield is larger for Pt/C than for MCS/C (Fig. S2), an effect that can uniquely be ascribed to the active layer thickness (H₂O₂ species cannot as easily diffuse out of thick active layers). The same can proceed in APEFC conditions, where thicker active layers in the case of MCS/C could better retain water. This is not necessarily a result of the catalyst's behaviour, but could also be due to the carbon substrate, that may be more oxophilic in the MCS/C case, because (i) Vulcan XC-72 has been preheated at 110°C in air, (ii) has undergone an overall oxidative treatment upon deposition of the MCS nanoparticles. The APEFC data with varying loading of MCS/C support this statement, as the experimental data obtained at twice smaller loading of MCS/C are significantly smaller than at 0.58 mg/cm² (Figure S3), leading to same active layer thickness than for the Pt/C nominal cathode, approximately, but much smaller performances (than the 0.58 mg/cm² MCS/C cathode and the nominal Pt/C cathode). So it seems that the performances do scale non-negligibly to the electrode morphology/thickness. So, the same tests shall be performed at fixed active layer thickness for Pt/C and for MCS/C (both in RDE, RRDE in in APEFC) to enable a fair comparison, and the results could be discussed as a function of the active layer's hydrophilicity/hydrophobicity. An idea would be to benchmark MCS/C 40wt% with Pt/C 40 wt% (also commercially-available) – then both the active layer thickness and the catalyst content would be kept identical, rendering more obvious comparisons.

3- The DFT and MD calculations have been performed on defect-free surfaces. However, it is not sure at all that these surfaces are "ideally-shaped" under electrochemical control. Many studies of the literature indeed reveal the progressive leaching of Mn (and Co) from such catalysts. So, have the authors observed their MCS/C catalysts after electrochemistry. Are they still presenting the same surface chemistry and texture as before? I suggest such tests are performed, which could be as simple as XRD and TEM post electrochemistry (not necessarily in situ, of course, ex situ would be fine). In addition, if they detected defects on their materials (and these could simply be e.g. low-coordination sites that are classically observed at NPs), what is the influence of these on the observed performances and overall ORR pathway/mechanism?

4- The in situ FTIR is very interesting and seems to support the authors' views (I am convinced, essentially). Nevertheless, again here, could there be an effect of the ionomer used (Nafion, I suspect,

this is not detailed) and of the active layer thickness (not detailed either) and the trends observed? Doubts would be easily ruled out by using similarly-loaded MCS/C and Pt/C as well as the alkaline ionomer as a binder.

5- "Although MCS-like materials were used as ORR catalysts in the literature, the cell performance has never reached such a high level as reported in this work, and the mechanistic understanding has been vague" is completely true: can the authors explain why they have managed to obtain such good performances with MCS/C in this study? How was the material differently handled than in ref. 9-11?

Minor comments:

1- Ref. S2 is not the first one about RRDE determination of the peroxide yield. Please cite instead [O. Antoine et al., Journal of Applied Electrochemistry 30 (2000) 839-844] for acidic electrolytes or [L. Genies et al., Electrochimica Acta 44 (1998) 1317-1327] for alkaline ones.

2- Why is the HFR always slightly higher for MCS/C cathodes than for Pt/C cathodes? Is it a result of the larger active layer thickness in the former case? If so, it again artificially improves the iR-corrected MCS/C performances versus Pt/C, and a fairer comparison would be made at fixed active layer thickness for Pt/C and MCS/C.

3- The format of the citation cited is not homogeneous. Please repair.

4- Some studies about durability of electrocatalysts and membranes in alkaline environment could be cited in the introduction: this fields starts to be explored.

End.

Reviewer #2 (Remarks to the Author):

The manuscript reports a very important finding about the ORR catalysis for alkaline polymer electrolyte fuel cells (APEFCs). Although quite a few non-Pt catalysts have been reported in the literature that are comparable to the commercial Pt catalyst toward the ORR in alkaline solutions, none has actually performed well in real APEFC tests. The Mn-Co spinel catalyst reported in this paper is the first one that exhibits comparable, or even better, performance than the Pt catalyst in the cathode of an APEFC. More importantly, the authors have also unraveled the underlying mechanism for the seemingly controversial results from the CV and fuel cell measurements, which have long been puzzling the research community in this area. The key discovery of this manuscript is that the outstanding performance of the Mn-Co catalyst results from the synergistic effect between the Mn sites binding O₂ and the Co sites activating H₂O, which facilitates the proton-coupled electron transfer processes of oxygen reduction. Such a synergistic mechanism has not been well recognized previously, and is highly insightful for the ORR catalyst design, for the APEFCs in particular.

Based on the above assessments, the reviewer thus strongly supports a quick publication of this important work, after the authors clarify the following points in their manuscript.

1) The Mn-Co spinel was a documented material for the ORR catalysis, but the catalytic activity was not as good as that reported in this manuscript. I would like to suggest that the authors discuss/explain this point in the text. Is it because of a structural reason or particle size effects, among others?

2) The electronic conductivity of oxide catalysts is usually not comparable to that of metal catalysts. How does it influence the catalytic activity of the Mn-Co spinel catalyst in this work?

3) The APEFC performance is very impressive, as demonstrated in Figs. 1b & 1c. Current densities as high as 3 A/cm² were delivered by either the Pt cathode or the MCS cathode. The authors have clarified in the text that the performance difference is not due to the electrical resistance of the electrode, but it should still be related to the thickness of the catalyst layer. It is suggested that the authors clarify whether such a structural factor had played a role in the cell performance.

4) In Fig. 4b, the stark effect can be clearly observed on MCS surface, which is a strong evidence for the water activation. The IR wavenumber corresponds to the bending vibration of the water molecule. Why was the stretching vibration not applied in this analysis, because the stretching vibration is supposed to be stronger than the bending vibration?

5) In light of the importance of the stability of the MCS catalyst, a brief comment/discussion on stability may be provided in the manuscript.

Wuhan University

Lin Zhuang
Chang Jiang Professor
Department of Chemistry
Wuhan 430072, China
Tel./Fax: +86 27 68753833
E-mail: lzhuang@whu.edu.cn
Web: www.zhuang.whu.edu.cn

March 11, 2019

Reviewer #1 (Remarks to the Author):

Review form for manuscript NCOMMS-18-33282-T “Synergistic Mn-Co Catalyst Outperforms Pt on High-Rate Oxygen Reduction for Alkaline Polymer Electrolyte Fuel Cells”, by Ying Wang et al. The paper is a very comprehensive study of carbon-supported mixed Co and Mn oxides (MCS/C) for the alkaline oxygen reduction reaction, for potential use (and validation) in Alkaline polymer electrolyte fuel cells (APEFCs). The materials demonstrate a clear interest, as their performance is matching (or even surpassing) those of a Pt/C benchmark. The experimental data contains thorough physicochemical characterizations of the materials, as well as theoretical calculations by DFT and molecular dynamics. This is a very nice set of data, that would deserve publication, provided the following questions are adequately taken care of by the authors.

Major comments:

1- In RDE, the binder/ionomer was Nafion, whereas in APEFC it was the alkaline ionomer (*a*QAPS-S₁₄); why did the authors make this choice? Could it influence the observed performances? For example, it starts to be observed that alkaline ionomers act as “poisons” of Pt electrocatalysts, while Nafion does less; if so, would the authors not “artificially” make their Pt/C electrode better in RDE than in APEFC? Please provide as well RDE data using the same alkaline ionomer for the two materials (Pt/C and MCS/C) to enable a fairer comparison between RDE and APEFC.

Thanks for the comment. The use of Nafion ionomer in RDE measurements is a tradition in the literature, be the electrolyte an acid or alkaline solution. We did have compared the Nafion ionomer with alkaline counterpart, *i.e.*, *a*QAPS-S₁₄ ionomers, in RDE measurements, and results (Fig. R1) shows essentially no difference. Therefore, the comparison between Pt/C and MCS/C in RDE measurements is reliable and the conclusion does not depend on the ionomer applied.

Fig. R1 RDE measurements in O_2 -saturated 1.0 M KOH solution using 40 wt% Pt/C (Johnson Matthey, $50 \mu\text{g}_{\text{Pt}} \text{cm}^{-2}$) and 40 wt% MCS/C ($72 \mu\text{g}_{\text{metal}} \text{cm}^{-2}$) with Nafion or aQAPS-S₁₄ ionomers, respectively. Scan rate = 5 mV s^{-1} . Rotation rate = 1600 rpm.

2- The results can also be biased because the active layers (both in RDE and in APEFC configuration) have not the same thicknesses (the MCS percentage on carbon is 40 wt% versus 60 wt% for Pt/C, and the loading at the RDE/APEFC cathode is larger than for Pt/C by 50%, which means that the active layer is roughly twice thicker for MCS/C than for Pt/C in all conditions). This can strongly influence the results. This is already witnessed in RRDE, where the H_2O_2 yield is larger for Pt/C than for MCS/C (Fig. S2), an effect that can uniquely be ascribed to the active layer thickness (H_2O_2 species cannot as easily diffuse out of thick active layers). The same can proceed in APEFC conditions, where thicker active layers in the case of MCS/C could better retain water. This is not necessarily a result of the catalyst's behaviour, but could also be due to the carbon substrate, that may be more oxophilic in the MCS/C case, because (i) Vulcan XC-72 has been preheated at 110°C in air, (ii) has undergone an overall oxidative treatment upon deposition of the MCS nanoparticles. The APEFC data with varying loading of MCS/C support this statement, as the experimental data obtained at twice smaller loading of MCS/C are significantly smaller than at 0.58 mg/cm^2 (Figure S3), leading to same active layer thickness than for the Pt/C nominal cathode, approximately, but much smaller performances (than the 0.58 mg/cm^2 MCS/C cathode and the nominal Pt/C cathode). So it seems that the performances do scale non-negligibly to the electrode morphology/thickness. So, the same tests shall be performed at fixed active layer thickness for Pt/C and for MCS/C (both in RDE, RRDE in in APEFC) to enable a fair comparison, and the results could be discussed as a function of the active layer's hydrophilicity/hydrophobicity. An idea would be to benchmark MCS/C 40wt% with Pt/C 40wt% (also commercially available) – then both the active layer thickness and the catalyst content would be kept identical, rendering more obvious comparisons.

We fully understand the reviewer's concern about the influence of the carbon substrate and the catalyst layer. (i) The preparation process of MCS/C did not change the surface property of the carbon support (Vulcan XC-72). The purpose of preheating Vulcan XC-72 at 110°C in air was to

remove adsorbed water in carbon substrate. The carbon substrate remained stable under such conditions, as confirmed by previous reports (such as *Carbon*, **2005**, 43, 179; *Chem. Mater.*, **2006**, 18, 1498). The MCS nanoparticles were deposited on the carbon substrate by using $\text{NH}_3 \cdot \text{H}_2\text{O}$ as precipitant and subsequently hydrothermal reaction at 150°C . This synthetic method can hardly turn the carbon substrate to be more oxophilic. The C 1s spectra of XPS (Fig. R2) clearly indicate no increase in the oxophilic content of the carbon support in Pt/C and MCS/C.

Fig. R2 Carbon 1s spectra of XPS for pristine XC-72, and that in Pt/C and MCS/C.

(ii) The catalyst loading of MCS/C in the APEFC cathodes was $2.0 \text{ mg}_{\text{catalyst}} \text{ cm}^{-2}$ ($0.58 \text{ mg}_{\text{metal}} \text{ cm}^{-2} + 1.2 \text{ mg}_{\text{C}} \text{ cm}^{-2}$), which was an optimal result and is practical for application; while the catalyst loading of Pt/C was $0.4 \text{ mg}_{\text{Pt}} \text{ cm}^{-2}$, a usual level applied in the literature. It is actually hard to compare Pt and nonprecious metal catalyst on a 'fair' basis, and nonprecious catalysts are practically allowed to use in a larger amount due to the low cost. To exclude the possible effect of the thickness of the catalyst layer, we have done additional RRDE and fuel-cell tests for Pt/C and MCS/C with the same carbon loading, which produced essentially the same thickness of the catalyst layer. Results are shown in Fig. R3. Although these data are slightly different from those presented in the manuscript, the conclusion does not change. The kinetic current densities at 0.85 V are very close to those in Fig. S6, indicating that the observed phenomenon is due to a catalytic effect, rather than an electrode thickness effect.

Fig. R3 Comparison of MCS/C and Pt/C, with the same catalyst layer thickness predominantly determined by the carbon content. (a) RRDE measurements in O₂-saturated 1.0 M KOH solution using 25 wt% Pt/C (50 $\mu\text{g}_{\text{Pt}} \text{ cm}^{-2}$ + 150 $\mu\text{g}_{\text{C}} \text{ cm}^{-2}$) and 40 wt% MCS/C (72 $\mu\text{g}_{\text{metal}} \text{ cm}^{-2}$ + 150 $\mu\text{g}_{\text{C}} \text{ cm}^{-2}$), respectively. Scan rate = 5 mV s^{-1} . Rotation rate = 1600 rpm. (b) The calculated H₂O₂ yield percentage (%). (c) APEFC tests with H₂ and O₂ at 100% RH. Anode catalyst: 60 wt% Pt-Ru/C (0.4 $\text{mg}_{\text{metal}} \text{ cm}^{-2}$). Cathode catalyst: 25 wt% Pt/C (0.4 $\text{mg}_{\text{Pt}} \text{ cm}^{-2}$ + 1.2 $\text{mg}_{\text{C}} \text{ cm}^{-2}$) or 40 wt% MCS/C (0.58 $\text{mg}_{\text{metal}} \text{ cm}^{-2}$ + 1.2 $\text{mg}_{\text{C}} \text{ cm}^{-2}$). Operation temperature = 60°C. Backpressure = 0.1 MPa. See fig. R10 for iR -corrected plots and impedance measurements. (d) Performance comparison: Kinetic current densities (j_k) at 0.85 V calculated from the RRDE data recorded in 1.0 M KOH solution, and the peak power density (PPD) resulting from APEFC tests.

3- The DFT and MD calculations have been performed on defect-free surfaces. However, it is not sure at all that these surfaces are “ideally-shaped” under electrochemical control. Many studies of the literature indeed reveal the progressive leaching of Mn (and Co) from such catalysts. So, have the authors **observed their MCS/C catalysts after electrochemistry**. Are they still presenting the same surface chemistry and texture as before? I suggest such tests are performed, which could be as simple as XRD and TEM post electrochemistry (not necessarily in situ, of course, ex situ would be fine). In addition, if they detected defects on their materials (and these could simply be e.g. low-coordination sites that are classically observed at NPs), what is the influence of these on the observed performances and overall ORR pathway/mechanism?

Following the suggestions of the reviewer, we have done STEM observations of the MCS/C catalyst after electrochemical tests (e.g., 1000 potential cycles, 0.6-1.0 V vs. RHE, 100 mV s^{-1} in O₂-sat. 1 M KOH), making use of the FEI-F-20 Tecnai STEM and Oxford X-Max detector at Cornell University. The HAADF-STEM imaging (Fig. R4) shows that, after ink making and

potential cycling, the catalyst particles did aggregate to some degree in some regions, but the spinel crystal remains perfectly unchanged, as indicated by the electron diffraction result (Fig. R5), and, on the basis of EDS analysis, there is essentially no change in both the Mn/Co ratio (Fig. R6) and the Mn/Co distribution inside a MCS particle (Fig. R7). These results manifest that the MCS catalyst itself is structurally stable under the electrochemical conditions.

Fig. R4 HAADF-STEM images of MCS/C as synthesized (**a, b**) and after 1000 potential cycles (**c, d**).

Fig. R5 Electron diffraction pattern of MCS/C after 1000 potential cycles.

MCS/C	Mn at. %	Co at. %
As-synthesized	47.0 ± 0.6	53.0 ± 0.6
After 1000 cycles	48.5 ± 0.8	51.5 ± 0.8

Fig. R6 EDS quantitative analysis of MCS/C as-synthesized and after 1000 potential cycles. Mn, Co at.% was calculated based on the Cliff-Lorimer equation (*J. Microsc.* **1975**, 103, 203). Relative error was defined as on standard deviation and analyzed based on measurements on five random regions in TEM grid.

Fig. R7 STEM-EDS elemental mapping of MCS/C after 1000 potential cycles. Three particles were observed.

4- The in situ FTIR is very interesting and seems to support the authors' views (I am convinced, essentially). Nevertheless, again here, could there be an effect of the ionomer used (Nafion, I suspect, this is not detailed) and of the active layer thickness (not detailed either) and the trends observed? Doubts would be easily ruled out by using similarly-loaded MCS/C and Pt/C as well as the alkaline ionomer as a binder.

The possible difference between Nafion and *a*QAPS ionomers on the catalysis has been ruled out in previous discussion (Fig. R1). For FTIR experiments, the *a*QAPS-S₁₄ ionomer is not a good choice, since the stretching vibration of the benzene ring in 1620-1450 cm⁻¹ can interfere the analysis of the H₂O vibration (Fig. R8). Moreover, the ATR measurement is only sensitive to a very thin layer above the gold substrate, and all IR signals were collected relevant to a background state (1.2 V). There is no reason that the ionomer or the thickness of the catalyst layer could influence the observation of this experiment, *i.e.*, the potential-dependent shift in the wavenumber of the surface H₂O vibration (Fig. 4b).

Fig. R8 FTIR spectra of Nafion and *a*QAPS.

5- “Although MCS-like materials were used as ORR catalysts in the literature, the cell performance has never reached such a high level as reported in this work, and the mechanistic understanding has been vague” is completely true: can the authors explain why they have managed to obtain such good performances with MCS/C in this study? How was the material differently handled than in ref. 9-11?

We think that a careful control of the surface composition of the MCS catalyst is a key factor. As shown in Fig. R9, the cell performance is highly sensitive to the surface composition of the MCS-like catalysts, which can be easily changed during the synthetic process, even changing the aging temperature before the hydrothermal reaction. We provide synthetic details in the Method section.

Fig. R9 APEFC tests with H_2 and O_2 at 100% RH. Anode catalyst: 60 wt% PtRu/C ($0.4 \text{ mg}_{\text{metal}} \text{ cm}^{-2}$). Cathode catalyst: 40 wt% MCS/C, Mn-MCS/C or Co-MCS/C ($0.58 \text{ mg}_{\text{metal}} \text{ cm}^{-2}$). Operation temperature = 60°C . Backpressure = 0.1 MPa.

Minor comments:

1- Ref. S2 is not the first one about RRDE determination of the peroxide yield. **Please cite instead** [O. Antoine et al., Journal of Applied Electrochemistry 30 (2000) 839-844] for acidic electrolytes or [L. Genies et al., Electrochimica Acta 44 (1998) 1317-1327] for alkaline ones.

Thanks for the comment. The relevant citation has been revised and appears as Ref. 36 in the main text.

2- Why is the HFR always slightly higher for MCS/C cathodes than for Pt/C cathodes? Is it a result of the larger active layer thickness in the former case? If so, it again artificially improves the iR -corrected MCS/C performances versus Pt/C, and a fairer comparison would be made at fixed active layer thickness for Pt/C and MCS/C.

The cell performance with the same cathode thickness of Pt/C and MCS/C has been presented in Fig. R3c. The iR -corrected performance and HFR are shown here in Fig. R10. While the HFR of the Pt/C cathode becomes closer to that of the MCS/C cathode, the iR -corrected performance of the Pt/C is still significantly lower than that of the MCS/C cathode. Therefore, the iR -corrected performance presented in the SI is reliable.

Fig. R10 Comparison between the Pt/C and MCS/C cathodes with the same thickness of catalyst layer. (a) *iR*-corrected cell performance. (b) High frequency resistance (HFR) recorded under different current densities.

3- The format of the citation cited is not homogeneous. Please repair.

Done.

4- Some studies about durability of electrocatalysts and membranes in alkaline environment could be cited in the introduction: this fields starts to be explored.

We have added more mentioned literatures in the introduction.

Reviewer #2 (Remarks to the Author):

The manuscript reports a very important finding about the ORR catalysis for alkaline polymer electrolyte fuel cells (APEFCs). Although quite a few non-Pt catalysts have been reported in the literature that are comparable to the commercial Pt catalyst toward the ORR in alkaline solutions, none has actually performed well in real APEFC tests. The Mn-Co spinel catalyst reported in this paper is the first one that exhibits comparable, or even better, performance than the Pt catalyst in the cathode of an APEFC. More importantly, the authors have also unraveled the underlying mechanism for the seemingly controversial results from the CV and fuel cell measurements, which have long been puzzling the research community in this area. The key discovery of this manuscript is that the outstanding performance of the Mn-Co catalyst results from the synergistic effect between the Mn sites binding O₂ and the Co sites activating H₂O, which facilitates the proton-coupled electron transfer

processes of oxygen reduction. Such a synergistic mechanism has not been well recognized previously, and is highly insightful for the ORR catalyst design, for the APEFCs in particular.

Based on the above assessments, the reviewer thus strongly supports a quick publication of this important work, after the authors clarify the following points in their manuscript.

1) The Mn-Co spinel was a documented material for the ORR catalysis, but the catalytic activity was not as good as that reported in this manuscript. I would like to suggest that the authors discuss/explain this point in the text. Is it because of a structural reason or particle size effects, among others?

As we have answered the similar comment from Reviewer #1, we find that the performance of the MCS catalyst is highly sensitive to the surface composition. The crystal structure is spinel, as reported in the literature, and the particle size of our sample is also close to those documented. Hence we think the surface composition is the key factor. We provide the preparation detail in the Method section.

2) The electronic conductivity of oxide catalysts is usually not comparable to that of metal catalysts. How does it influence the catalytic activity of the Mn-Co spinel catalyst in this work? Based on the structural characterization (Fig. 2), the stoichiometry of the MCS was determined to be $[\text{Mn}_{0.3}\text{Co}_{0.7}][\text{Mn}_{0.6}\text{Co}_{0.4}]_2\text{O}_4$, so it is electronically conductive through the hopping of charge between the mixed arrangement atoms in octahedral sites (*J. Am. Ceram. Soc.*, **2007**, 90, 1515). Furthermore, after depositing on carbon substrate, the electronic conductivity of the MCS/C is not a problem, as can be judged from the slope of the I-V curves at middle polarization region (Fig. 1b & 1c).

3) The APEFC performance is very impressive, as demonstrated in Figs. 1b & 1c. Current densities as high as 3 A/cm² were delivered by either the Pt cathode or the MCS cathode. The authors have clarified in the text that the performance difference is not due to the electrical resistance of the electrode, but it should still be related to the thickness of the catalyst layer. It is suggested that the authors clarify whether such a structural factor had played a role in the cell performance.

We have clarified this point in answering the similar comment from Reviewer #1 and demonstrated cell performance based on the same cathode thickness (Fig. R3). Clearly, the observed phenomenon, as well as the superiority of the MCS cathode, is not due to the slight difference in the thickness of the catalyst layer.

4) In Fig. 4b, the stark effect can be clearly observed on MCS surface, which is a strong evidence for the water activation. The IR wavenumber corresponds to the bending vibration of the water molecule. Why was the stretching vibration not applied in this analysis, because the stretching vibration is supposed to be stronger than the bending vibration?

The O-H stretching vibrations of H₂O was overlapped by that of the OH⁻, in the range of wavenumber 3700-3200 cm⁻¹, hence the bending vibration signal of H₂O was chosen so as to avoid the interference from OH⁻.

5) In light of the importance of the stability of the MCS catalyst, a brief comment/discussion on stability may be provided in the manuscript.

Thanks for the comments. We add a stability result of MCS/C (10000 potential cycles in O₂-saturated 1.0 M KOH solution) in SI and a comment at the end of the main text. As shown in Fig. R11, only 10 mV negative shift in the half-wave potential was observed, indicating a good stability of the MCS catalyst for the ORR.

Fig. R11 Stability test of MCS in O₂-saturated 1.0 M KOH solution. The stability was evaluated under 10000 potential cycles from 0.6 V to 1.0 V at 0.1 V s⁻¹. The rotation rate of RDE test was 1600 rpm, and the scan rate was 5 mV s⁻¹.

REVIEWERS' COMMENTS:

Reviewer #1 (Remarks to the Author):

The answers of the authors to my comments are fully satisfactory. I strongly recommend publication of this nice work.

Reviewer #2 (Remarks to the Author):

The authors have satisfactorily addressed the points we raised in the previous round of review and I feel that the paper is now acceptable for publication.

Reviewer #1 (Remarks to the Author):

The answers of the authors to my comments are fully satisfactory. I strongly recommend publication of this nice work.

Reviewer #2 (Remarks to the Author):

The authors have satisfactorily addressed the points we raised in the previous round of review and I feel that the paper is now acceptable for publication.

We are happy to see the strong support from the reviewers!